

# Genomic organization and expression profiles of nitrogen assimilation genes in *Glycine max*

Hind Abdelmonim Elsanosi[1,2], Tiantian Zhu[1], Guisheng Zhou[1] and Li Song[1]

[1] Joint International Research Laboratory of Agriculture and Agri-Product Safety, the Ministry of Education of China, Jiangsu Key Laboratory of Crop Genomics and Molecular Breeding, Yangzhou University, Yangzhou, Jiangsu, China
[2] Faculty of Agriculture, University of Khartoum, Khartoum, Sudan

## ABSTRACT

**Background:** Glutamine synthetase (GS), glutamate synthase (GOGAT), and nitrate reductase (NR) are key enzymes involved in nitrogen assimilation and metabolism in plants. However, the systematic analysis of these gene families lacked reports in soybean (*Glycine max* (L.) Merr.), one of the most important crops worldwide.
**Methods:** In this study, we performed genome-wide identification and characterization of *GS*, *GOGAT*, and *NR* genes in soybean under abiotic and nitrogen stress conditions.
**Results:** We identified a total of 10 *GS* genes, six *GOGAT* genes, and four *NR* genes in the soybean genome. Phylogenetic analysis revealed the presence of multiple isoforms for each gene family, indicating their functional diversification. The distribution of these genes on soybean chromosomes was uneven, with segmental duplication events contributing to their expansion. Within the nitrogen assimilation genes (NAGs) group, there was uniformity in the exon-intron structure and the presence of conserved motifs in NAGs. Furthermore, analysis of cis-elements in NAG promoters indicated complex regulation of their expression. RT-qPCR analysis of seven soybean NAGs under various abiotic stresses, including nitrogen deficiency, drought-nitrogen, and salinity, revealed distinct regulatory patterns. Most NAGs exhibited up-regulation under nitrogen stress, while diverse expression patterns were observed under salt and drought-nitrogen stress, indicating their crucial role in nitrogen assimilation and abiotic stress tolerance. These findings offer valuable insights into the genomic organization and expression profiles of *GS*, *GOGAT*, and *NR* genes in soybean under nitrogen and abiotic stress conditions. The results have potential applications in the development of stress-resistant soybean varieties through genetic engineering and breeding.

# INTRODUCTION

Nitrogen is an essential nutrient for plant growth and development, playing a key role in various physiological processes such as protein synthesis, nucleic acid production, and regulation of enzyme activity. The availability of nitrogen greatly influences crop

Corresponding author
Li Song, songli@yzu.edu.cn

productivity (*Gaudinier et al., 2018*). Furthermore, nitrogen metabolism in plants is a complex process that involves various physiological and biochemical steps, including uptake, transportation, assimilation, and mobilization, all of which are essential for protein synthesis. Plants primarily absorb inorganic nitrogen in the forms of nitrate ($NO_3^-$) and ammonium ($NH_4^+$) from the soil (*Ladha et al., 1998*; *Li et al., 2022*). Nitrate transporters (NRT) and ammonium transporters (AMT) play key roles in facilitating this active uptake process. During nitrogen assimilation, $NO_3^-$ is converted to $NH_4^+$ *via* the main nitrate assimilation pathway, catalyzed by key enzymes such as nitrate reductase (NR) and nitrite reductase (NiR). Within plants, nitrogen is further incorporated into amino acids through the glutamine synthetase/glutamate synthase (GS/GOGAT) cycle, which is essential for protein synthesis (*Konishi & Yanagisawa, 2014*). The efficiency of nitrogen use (NUE) is a critical trait for crops, which is influenced by nitrogen absorption and transport processes. Efficient nitrogen uptake and rational transport are predominantly mediated by membrane-localized transporters to optimize NUE (*Liu, Hu & Chu, 2022*; *Zhao et al., 2022*).

Glutamine synthetase (GS), glutamate synthase (GOGAT), and nitrate reductase (NR) are essential and interrelated genes in plant nitrogen metabolism, playing key roles in nitrogen assimilation, remobilization, storage, reutilization and stress resistance (*Liu et al., 2016*; *Sajjad et al., 2021*). Nitrate reductase (NR) is one of the key enzymes for plant nitrogen assimilation and root architecture remodeling (*Fu et al., 2020*). NR is essential for converting nitrate ($NO_3^-$) to nitrite ($NO_2^-$), initiating the process of inorganic nitrogen utilization (*Tang et al., 2022*).

GS and GOGAT play a crucial role in converting inorganic ammonium salts to organic nitrogen compounds through the GS/GOGAT cycle in roots (*García-Gutiérrez, Cánovas & Ávila, 2018*; *Lebedev et al., 2018*). GS facilitates the conversion of $NH_4^+$ and glutamate (Glu) into glutamine, with glutamine further catalyzed back to Glu by glutamate synthase (GOGAT) to participate in the amino acid cycle (*Zhong et al., 2019*). In higher plants, two types of GS are presented with distinct subcellular localization: cytosolic GS1 and chloroplastic GS2 (*Yin et al., 2022*). GS1 plays a crucial role in the initial assimilation of $NH_4^+$ or $NO_3^-$ from the soil and also participates in nitrogen assimilation from processes releasing $NH_4^+$, such as amino acid transamination and protein degradation during senescence (*Thomsen et al., 2014*; *Krapp, 2015*; *Fortunato et al., 2023*). Conversely, GS2 is predominantly responsible for assimilating $NH_4^+$ generated during photorespiration in chloroplasts (*Daniel-Vedele & Chaillou, 2005*; *Masclaux-Daubresse et al., 2010*; *Rashid et al., 2022*). The expression of *GS* gene is intricately regulated by factors such as light exposure, nitrogen availability, and the plant's developmental stage (*Fortunato et al., 2023*; *Larios et al., 2004*).

Glutamate synthase (GOGAT) plays a pivotal role in the conversion of glutamine and 2-oxoglutarate into two glutamate molecules. One of these glutamate molecules is vital for the synthesis of nitrogen-containing compounds such as amino acids, nucleotides, and chlorophyll, while the other serves as a substrate for glutamine synthetase (GS) to initiate the GS/GOGAT cycle (*Suzuki, 2021*). GOGAT exists in two forms: ferredoxin-dependent GOGAT (Fd-GOGAT) located in chloroplasts and plastids and NADH-dependent

GOGAT (NADH-GOGAT) in the cytoplasm (*Yin et al., 2022*). Fd-GOGAT can couple with GS in chloroplasts to catalyze the assimilation of ammonium released during photorespiration (*Liu et al., 2016*).

Soil salinity, drought, and high temperatures are common environmental stresses that have a significant impact on plant growth and global crop yield (*Zang et al., 2018*). Under stress conditions, plants activate specific nitrogen assimilation genes (NAGs) to improve nitrogen uptake, assimilation, and remobilization. This adaptive response helps plants efficiently utilize nitrogen, allowing them to survive and thrive in challenging conditions. Numerous studies have emphasized the role of NAGs in plant reactions to abiotic and nitrogen stresses. For instance, increasing *GS* genes in rice have been shown to provide resistance to salt, drought and cold stress (*Cai et al., 2009*), while introducing pine cytoplasmic glutamine synthetase (GS1) into transgenic poplar has enhanced tolerance to drought stress (*El-Khatib et al., 2004*). Additionally, the induction of *TaGS2* expression by $NO_3^-$ was observed in wheat leaves. In *Zostera marina* L., the expression of the *NR* gene increased in response to NaCl treatment (*Lv et al., 2018*).

Soybean (*Glycine max* (L.) Merr.) is a highly valued crop known for its economic and nutritional benefits, including its high oil content (18%) and quality proteins (~40%), as well as positive effects on soil fertility, productivity, and profitability. It is often referred to as a miracle crop (*Modgil et al., 2021*; *Sun et al., 2023*; *Zhan et al., 2020*). In soybean plants, nitrogen-fixing rhizobial bacteria in root nodules provide a significant portion of the plant's nitrogen needs over its life cycle. However, nodule formation is a time-consuming process, and soybean seedlings initially depend on soil nitrogen (*Dai et al., 2021*; *Gan et al., 2003*). Applying a small amount of nitrogen fertilizer during the seedling stage can enhance photosynthesis, promote root growth, and facilitate rhizobia infection and nodule formation. Nevertheless, excessive or mistimed nitrogen application can impede root system growth and nitrogen fixation, leading to decreased yields (*Cafaro La Menza et al., 2020*). Environmental stressors such as drought, salinity, extreme temperatures, and nitrogen stress can significantly impact the growth of soybean plant, leading to reduced yields and cultivation challenges (*Bu et al., 2023*; *Gavili, Moosavi & Haghighi, 2019*). These stressors induce disruptions at the cellular level, resulting in secondary challenges such as the production of reactive oxygen species (ROS), cell membrane damage, protein misfolding, and osmotic stress within cells (*Staniak, Szpunar-Krok & Kocira, 2023*). Moreover, they impact water and nutrient uptake, resulting in decreased seed germination and impaired vegetative growth (*Shu et al., 2017*). During the reproductive stage, these stressors influence the number of flowers, pods, and seeds per soybean plant. Additionally, they modify the activity of diverse metabolites in soybeans, leading to reduced levels of carbohydrates, lipids, proteins, and secondary metabolites, ultimately affecting the quality and quantity of oil and protein contents. Consequently, this can lead to decreased yield and quality of soybean (*Das, Rushton & Rohila, 2017*).

The *GS*, *GOGAT*, and *NR* gene families have been studied in other plant species, like rice, pecan, and rapeseed (*He et al., 2021*; *Li et al., 2022*; *Qiao et al., 2023*), a comprehensive genome-wide analysis of these gene families in soybean is still lacking. This study aims to fill this gap by identifying and analyzing these gene families in six legume species: *Glycine*

*max*, *Cicer arietinum*, *Lotus japonicus*, *Phaseolus lunatus*, *Phaseolus vulgaris*, and *Vigna unguiculata*, with chromosome numbers of 20, 8, 6, 11, 11, and 11, respectively. Various analysis was conducted, including phylogenetic tree construction, gene structure and motif analysis, chromosomal location analysis, gene duplication, synteny analysis, cis-element identification, and protein structure prediction specifically in soybean genes. Additionally, the response of these genes to abiotic and nitrogen stresses was analyzed using RT-qPCR. This research will enhance our understanding of nitrogen utilization and stress tolerance in soybean, providing valuable insights for crop improvement strategies.

## MATERIALS AND METHODS

### Identification of NAGs in *Glycine max*

The amino acid sequences of *Arabidopsis GS*, *GOGAT*, and *NR* genes were retrieved from the *Arabidopsis* Information Resource (TAIR) database. These sequences were used as queries for a BLASTP search against the *Glycine max* reference genome (a4.v1 version) to identify members of the soybean *GS*, *GOGAT*, and *NR* gene families. Subsequently, domain analysis tools such as CD search (http://www.ncbi.nlm.nih.gov/Structure/cdd/wrpsb.cgi), PFAM (http://pfam.xfam.org/), and SMART (http://smart.embl-heidelberg.de/) were employed with default cut-off parameters were used to validate the accuracy of these genes. All candidate genes were aligned with *Arabidopsis* homologous genes. Genomic DNA, cDNA, CDS, and protein sequences of the *GS*, *GOGAT*, and *NR* genes were obtained from Phytozome. The same methods were applied to identify the *GS*, *GOGAT*, and *NR* genes in the other five legumes species (*C. arietinum*, *L. japonicas*, *P. lunatus*, *P. vulgaris*, and *V. unguiculata*). The confirmed novel NAGs genes were then renamed using a combination of the species abbreviation and the chromosome position.

### Analysis of physicochemical properties

Amino acid properties and physicochemical traits, including molecular weight (MW), aliphatic index, instability index (II), and isoelectric point (pI) of *GmGS*, *GmGOGAT*, and *GmNR* proteins were calculated using the ProtParam tool (https://web.expasy.org/protparam) (*Gasteiger et al., 2005*). Subcellular localization was predicted using an advanced protein prediction tool WOLF PSORT (https://wolfpsort.hgc.jp/) (*Horton et al., 2007*).

### Phylogenetic relationship and sequence alignment

The protein sequences of *G. max*, *C. arietinum*, *L. japonicas*, *P. lunatus*, *P. vulgaris*, and *V. unguiculata* were aligned using the MUSCLE tool with default settings (*Edgar, 2004*). Evolutionary relationships of the NAGs were illustrated through neighbor-joining (NJ) trees for each gene family, constructed with MEGA 11 software using 1,000 bootstraps (*Tamura, Stecher & Kumar, 2021*). The resulting trees in Newick format were visualized with iTOL v4 (http://itol.embl.de/) (*Zhou et al., 2023*).
## Analysis of conserved motifs and gene structure

The motif-based sequence analysis tool MEME (https://meme-suite.org/meme/db/motifs) (*Bailey et al., 2015*) was used to predict the conserved motifs of each protein. Furthermore, details regarding the distribution of exons, introns, and coding sequences were extracted from the GFF3 files of soybean genome annotation data. The gene architectures were visualized using the TBtools software (*Chen et al., 2020*).

## Chromosome localization, gene duplication, and syntenic analysis of soybean NAGs

The *GS*, *GOGAT*, and *NR* genes were mapped to specific chromosomes of *G. max* by comparing their physical distances using GFF3 genome files from the Phytozome v13 database (https://phytozome-next.jgi.doe.gov/) (*Goodstein et al., 2012*). Gene position on the chromosomes was visualized with TBtools software. Collinearity and gene duplication events were examined and presented using the Multiple Collinearity Scan toolkit (MCScanX) with default settings. The collinearity between the homologous gene pairs was visualized using the Circos tool in TBtools. In order to explore the mechanism behind the amplification of NAGs, gene synteny analysis was conducted between *G. max* and *C. arietinum*, *G. max* and *P. acutifolius*, and *G. max* and *A. thaliana*, and the syntenic relationships were visualized using TBtools software.

## Selection pressure and promoter analysis of soybean NAGs

The Ka/Ks ratio was estimated using TBtools software to analyze the selection pressure among soybean NAGs genes within the *G. max* genome. Additionally, we examined the cis-regulatory elements in the promoter regions of NAGs by analyzing the upstream sequences (1,500) of NAG proteins downloaded from Phytozome through the PlantCARE database (https://bioinformatics.psb.ugent.be/webtools/plantcare/html/) (*Rombauts et al., 1999*). The distribution of putative cis-elements was visualized using TBtools.

## Gene expression pattern of the NAGs in soybean tissues

To examine the expression patterns of NAGs, we analyzed the FPKM (fragments per kilobase of transcript per million fragments mapped) values obtained from Phytozome across eight different tissues: root, root tip, lateral root, stem, leaf, shoot tip, open flower, and unopened flower. Further analysis of these expression patterns was carried out using the heatmap function in TBtools.

## Protein-protein interaction network

To investigate the interactions among soybean NAG proteins, the protein sequences were submitted to STRING V12 (https://string-db.org/) (*Szklarczyk et al., 2023*). The resulting protein-protein interaction (PPI) networks were then visualized using Cytoscape v3.10.1 (*Shannon et al., 2003*).

## Plant materials and treatments

Williams 82 seeds with uniform size were sterilized using a chlorine gas method (*Paz et al., 2006*), and were germinated in a hydroponic system under controlled conditions in a growth

chamber with a 16-h light and 8-h dark cycle at 25 °C. Once the seedlings reached the V1 developmental stage, they were moved to a modified MS liquid medium to assess the impact of different nitrate levels-high (54.3 mM $NO_3^-$-HN), normal (18.81 mM $NO_3^-$-NN), and low (6.27 mM $NO_3^-$-LN) concentrations-for a duration of 7 days (*Dai et al., 2021*; *Guo et al., 2021*). To impose drought-nitrate (D-N) stress, the seedlings were subjected to the specified above nitrate concentrations for 5 days after that the 15% PEG6000 was added to each nitrate treatments over 2 days (*Yang et al., 2023*). Furthermore, for salt stress experiments, the seedlings were exposed to 150 mM NaCl for 24 and 48 h (*Guo et al., 2023*). The control treatment exclusively utilized MS medium. Following each treatment, the roots of five plants from three separate biological replicates were harvested and quickly frozen in liquid nitrogen. The samples were ground into powder using a sterilized mortar and pestle in liquid nitrogen. The powdered samples were promptly transferred into 1.5 ml RNase-free micro tubes (Corning Incorporated, Corning, Jiangsu province, China) and stored at −80 °C.

## Total RNA extraction and qRT-PCR analysis

Total RNA was isolated from root using the RNA Pure Plant Kit (DNase1) (Cat#CW0559S; CWBIO, Taizhou, Jiangsu, China). The quality of the RNA samples was assessed for degradation or contamination by 1% agarose gel electrophoresis. Additionally, the purity (A260/A280 ratio) and concentration of the RNA samples were determined using a Nanodrop ND-1000 spectrophotometer (V3.7.9). The primers for quantitative real-time PCR (qRT-PCR) were designed using the IDT online software (https://sg.idtdna.com/) (Table S1). Specificity screening was performed using Phytozome BLAST with the *Glycine max* Wm82.a4.v1 genome as the reference.

To generate cDNA, 1 µg of total RNA was reverse transcribed utilizing HiScript III RT SuperMix for qPCR (Cat# R323-01; Vazyme, Nanjing, Jiangsu, China) following the manufacturer's instructions, reverse transcription reactions are performed at 50 °C for 15 min following by 85 °C for 5 s. The qRT-PCR was performed on a CFX96 real-time PCR system (Bio-Rad, Hercules, CA, USA) using ChamQ SYBR qPCR Master Mix (Cat# Q311, Vazyme Nanjing, Jiangsu, China) which includes SYBR Green1 and other components as specified by the manufacturer. The quantification was performed in triplicate with 10 µl reactions containing 5 µl of 2x ChamQ SYBR qPCR Master mix, 1 µl of primer 10 µM (forward + reversed), 3 µl RNase-free water, and 1 µl cDNA. The PCR conditions involved an initial denaturation at 95 °C for 30 s, followed by 40 cycles of 95 °C for 5 s, and 60 °C for 30 s. A melting curve analysis was conducted by gradually increasing the temperature to 95 °C (increment rates of 0.5 °C/s) after cooling to 65 °C for 5 s.

The raw quantification cycle (Cq) values for each reaction were generated by the Bio-Rad CFX Maestro (version 4.1) as shown in Table S2. The relative expression of the target genes was normalized to the housekeeping gene *Actin11* (Glyma18g290800) and calculated using the $2^{-\Delta\Delta Ct}$ method. Ct values were obtained from three biological replicates, each with three technical replicates. Statistically significant differences in gene expression were determined using a t-test in Excel. Additional qPCR specifics are provided in a MIQE checklist table (Table S3).

## RESULTS

### Identification and phylogenetic analysis of soybean NAGs

Utilizing bioinformatics techniques, we identified 20, 9, 7, 11, 10, and 10 NAGs in the entire *G. max*, *C. arietinum*, *L. japonicas*, *P. lunatus*, *P. vulgaris*, and *V. unguiculata* genomes, respectively. ProtParam tool analysis using the ProtParam tool revealed significant differences in molecular weights of GS (ranging from 17.23 to 92.67 kDa), GOGAT (177.20 to 482.52 kDa), and NR (98.27 to 100.08 kDa) proteins in *G. max*. The amino acid sequence length varied from 155 to 4,395 bp, with a notable diversity in genes encoding GS, GOGAT, and NR. The pI values of all NAGs were less than 7 except *GmGS1*, indicating acidic nature of these proteins. Most soybean NAG proteins exhibited an instability index (II) value below 40, suggesting their stability. Subcellular location predications suggested that the majority of the NAGs are located in the cytoplasm and chloroplast, with a few genes also present in the mitochondria and nucleus, as shown in Table S4.

The phylogenetic relationships of NAGs in six legumes, including soybean, were investigated using the neighbor joining method. The *GS* genes were divided into four groups based on the phylogenetic tree, with groups 1 and 2 consisting of cytoplasmic *GS* genes, and groups 3 and 4 containing chloroplast *GS* genes (Fig. 1A). The phylogenetic analysis revealed that GOGAT family genes could be classified into three groups with group A consisting 7 *GOGAT* genes, all from the Fd-GOGAT subfamily and groups B and C containing Fd-GOGAT subfamily genes from various plants (Fig. 1B). Additionally, NR genes were categorized into three groups (Fig. 1C).

### Chromosomal location and gene duplication analysis of soybean NAGs

The distribution of the 20 NAGs in soybean was uneven across 13 out of the 20 soybean chromosomes. Chromosome 14 was notable for containing four genes, while the other chromosomes generally contained one or two genes each. To investigate the role of gene duplication in the expansion of soybean NAGs, we conducted an annotation and analysis of the intraspecific collinearity of these genes. Our analysis revealed that within the *GmGS*, *GmGOGAT*, and *GmNR* genes, there were 14, four, and three segmental duplications, respectively (Fig. 2). Additionally, we identified two tandem duplications. One was located on chromosome Gm02 (*GmGS1*/*GmGS2*), and the other on chromosome Gm14 (*GmNIA3*/*GmNIA4*). These tandem duplications involved genes from the same family and were positioned very close to each other on their respective chromosomes (Fig. S1).

Tandemly duplicated NAGs, like *GmNIA3* and *GmNIA4*, were observed to cluster together in the phylogenetic tree (Fig. 1), suggesting a strong evolutionary link between these duplicated genes. The existence of segmental and tandem duplications underscores the importance of these mechanisms in shaping the genetic makeup of soybean, potentially influencing its adaptation and functional diversity in nitrogen assimilation processes. To explore whether selective constraints influenced the duplicated genes, we analyzed the Ka/Ks ratio using the full-length protein sequences of the NAGs. The pairwise comparison

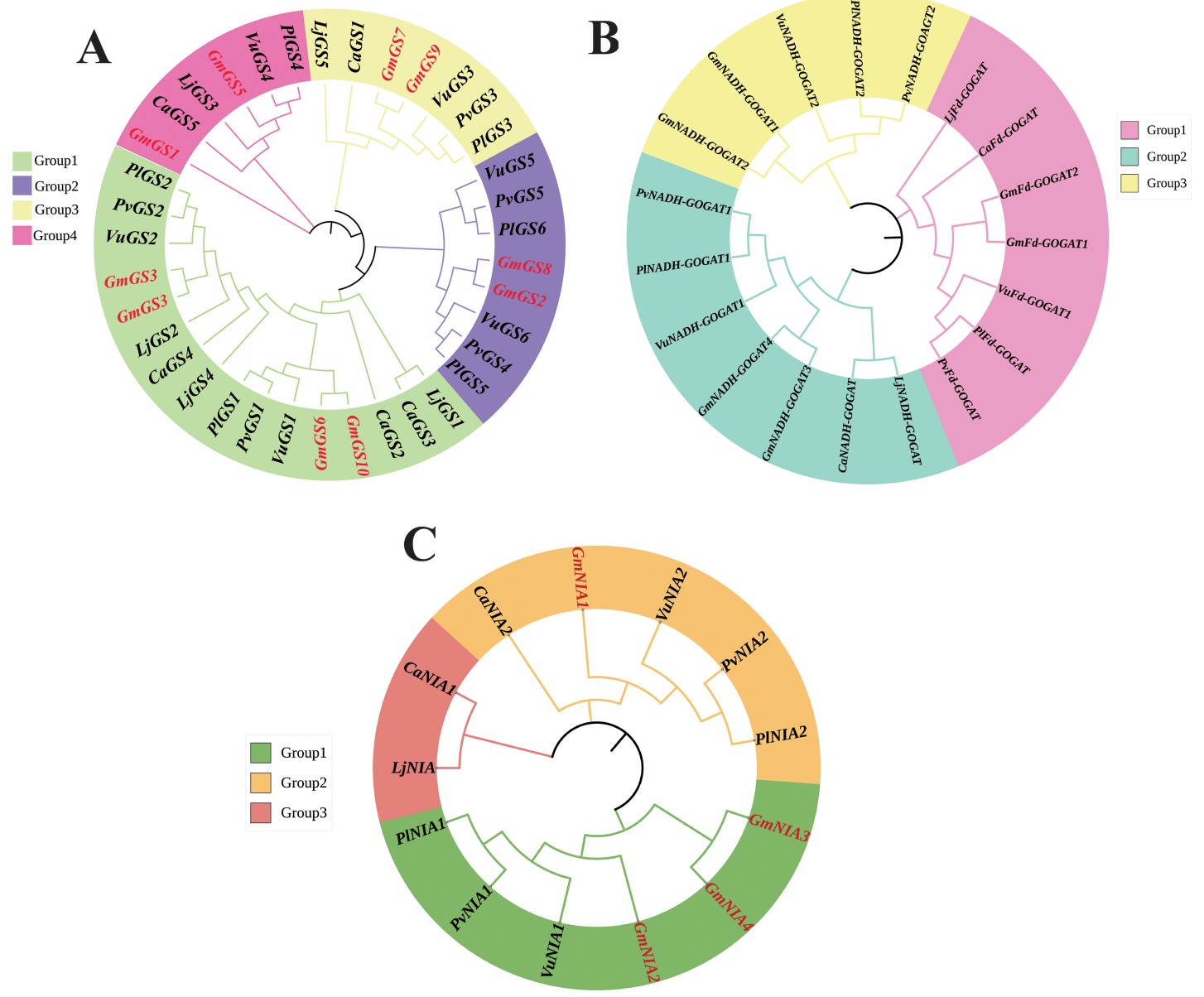

**Figure 1** Unrooted phylogenetic tree of *GS* genes (A) *GOGAT* genes (B), and *NR* genes (C) in *G. max*, *C. arietinum*, *L. japonicas*, *P. lunatus*, *P. vulgaris*, and *V. unguiculata*. The deduced full-length amino acid sequences were utilized to construct the phylogenetic tree using MEGA 11 software through a neighbor-joining method with 1,000 bootstrap replicates. Various groups are distinguished by different colors.

revealed a Ka/Ks ratio range of 0.04–0.19, which is notably less than 1, indicating that the soybean NAGs underwent purifying selection pressure with limited functional divergence. Moreover, the average Ka/Ks value for the *GS* gene family members was lower than that of the *GOGAT* and *NR* gene families, implying a slower evolution of *GmGS*s and highlighting their higher conservation level (Table S5).

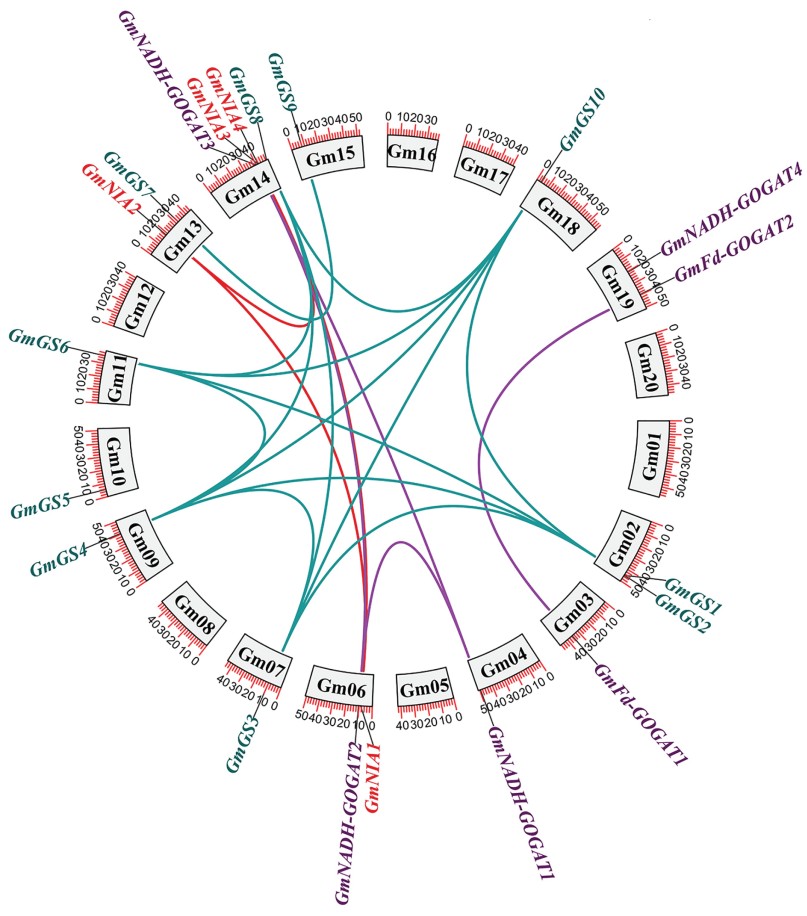

**Figure 2 Genomic distribution and duplication of the *GmGS*, *GmGOGAT*, and *GmNR* genes across 20 chromosomes of soybean.** The colorful lines indicate duplicated *GmGS*, *GmGOGAT*, and *GmNR* genes pairs.

## Structure of genes and conserved motifs in NAGs encoded proteins

The structures of the *GmGS*, *GmGOGAT*, and *GmNR* genes are illustrated in Fig. 3 to emphasize their structural diversity. *GmGS* genes exhibit significant variation in the number and length of introns, with *GmGS1* having five introns and *GmGS5* containing 17 introns, while other *GmGS* genes typically have 11–13 introns (Fig. 3A). The arrangement of introns differs among phylogenetic groups, enhancing the structural and functional diversity of *GmGS* genes. Moreover, cytoplasmic *GmGS* genes feature two types of introns (phase-0, phase-1), whereas chloroplastic *GmGS* genes contain phase-0, phase-1, and phase-2 introns, except for *GmGS1*.

In contrast, the distribution of introns in *GmGOGAT* genes reveals that all *GmNADH-GOGAT* genes have 21 introns, whereas *GmFd-GOGAT* genes have 32 introns with three different phase types (Fig. 3B). The *GmNR* genes, have a relatively small number of introns compared to the *GS* and *GOGAT* gene families, typically possessing three to four introns with phase-0 and phase-2 introns (Fig. 3C).

In addition, the conserved motifs of the NAGs in soybean were identified based on their amino acid sequences using MEME software. Each family had 10 motifs identified. Genes

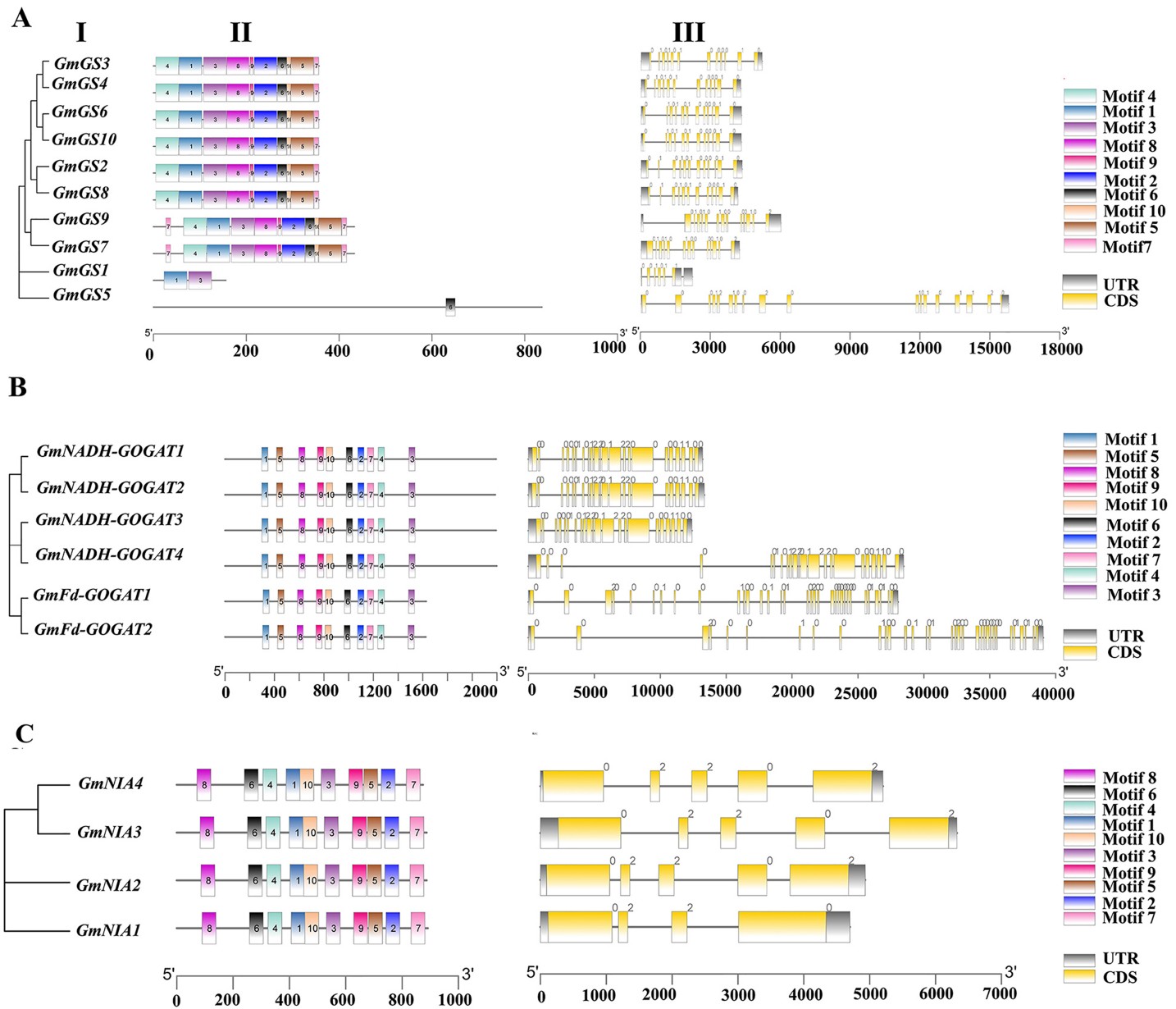

**Figure 3** **(I) Phylogenetic relationships, (II) motif compositions, (III) gene structure of *GmGS* (A), *GmGOGAT* (B), and *GmNR* (C).** Different colored boxes represent different motifs. The gray boxes represent UTR and the yellow boxes represent exons. In terms of introns, a phase 0 intron does not disrupt a codon, a phase1 disrupts a codon between the first and second bases, and the phase 2 intron located after second nucleotide of a codon.

with close phylogenetic relationships exhibited high similarity conserved motif composition. As shown in Fig. 3B, all genes in GOGAT family contained the 10 motifs, implies that this family may exhibit highly conserved functions or possibly functional redundancy among its genes, similar to the NR gene family (Fig. 3C). Notably, the *GS* gene family displayed some variation in motif arrangement, with *GmGS1* and *GmGS5* deviating from the pattern. *GmGS1* had Motif 1 and Motif 3, while *GmGS5* only had Motif 6. These findings suggest functional divergence among nitrogen assimilation genes in soybean.

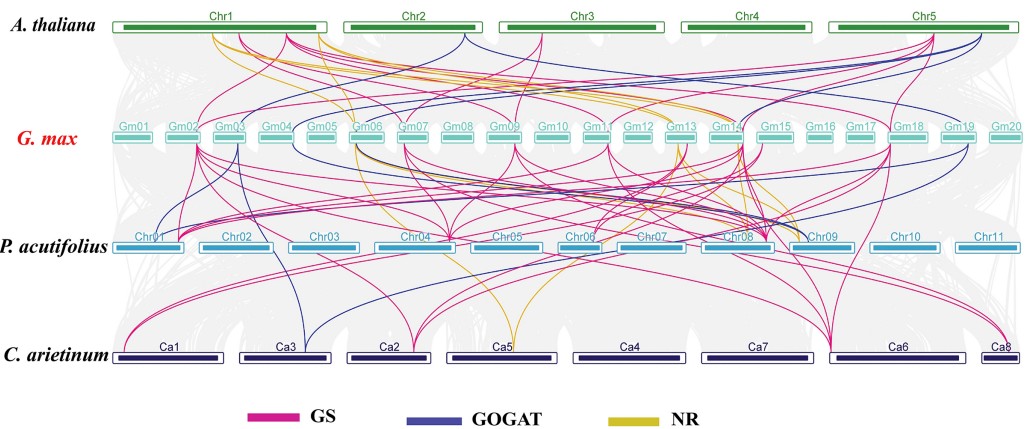

**Figure 4 Synteny analysis of NAGs between *Glycine max* and *A. thaliana*, *C. arietinum*, and *P. acutifolius*.** The gray lines in the background represent the collinear blocks within the soybean and other plant genomes. Pink lines highlight syntenic glutamine synthetase (*GS*) gene pairs, blue lines indicate syntenic glutamate synthase (*GOGAT*) gene pairs, and yellow lines represent syntenic nitrate reductase (*NR*) genes pairs.

## Synteny analysis of soybean NAGs

In order to investigate the evolutionary relationships of *GS*, *GOGAT*, and *NR* gene families across different species, we carried out an interspecies collinearity analysis involving *G. max*, *A. thaliana*, *C. arietinum*, and *P. acutifolius* (Fig. 4). The *GmGS* family members showed the highest number of collinear pairs with *P. acutifolius*, totaling 16 pairs, indicating a close evolutionary relationship between these two species. Additionally, five collinear gene pairs were identified between the *GmGOGAT* and *AtGOGAT* genes. Conversely, the syntenic relationships of *GmNR* genes with genes from other species were predominantly observed on two or three chromosomes. These results show that the presence of multiple collinear gene pairs among the three species was inferred to be genetic copies with lineage-specific amplification.

## Analysis of cis-acting elements in the promoter regions of the soybean NAGs

An analysis was conducted on the 1,500 bp upstream of the transcription start site of *GS*, *GOGAT*, and *NR* genes in soybean using the PlantCARE database (Fig. 5). The study revealed that these promoter regions contain three main types of cis-acting elements: light-responsive elements, hormone-responsive elements, and stress-responsive elements. Additionally, five types of hormone-responsive elements were identified, including gibberellin, abscisic acid, auxin, salicylic acid and jasmonic acid-responsive elements. Functional elements related to stress, such as low-temperature responsive elements, were also found. These findings indicate that the *GmGS*, *GmGOGAT*, and *GmNR* genes likely play crucial roles in various physiological processes in soybeans, such as plant growth, development, and responses to different stresses.

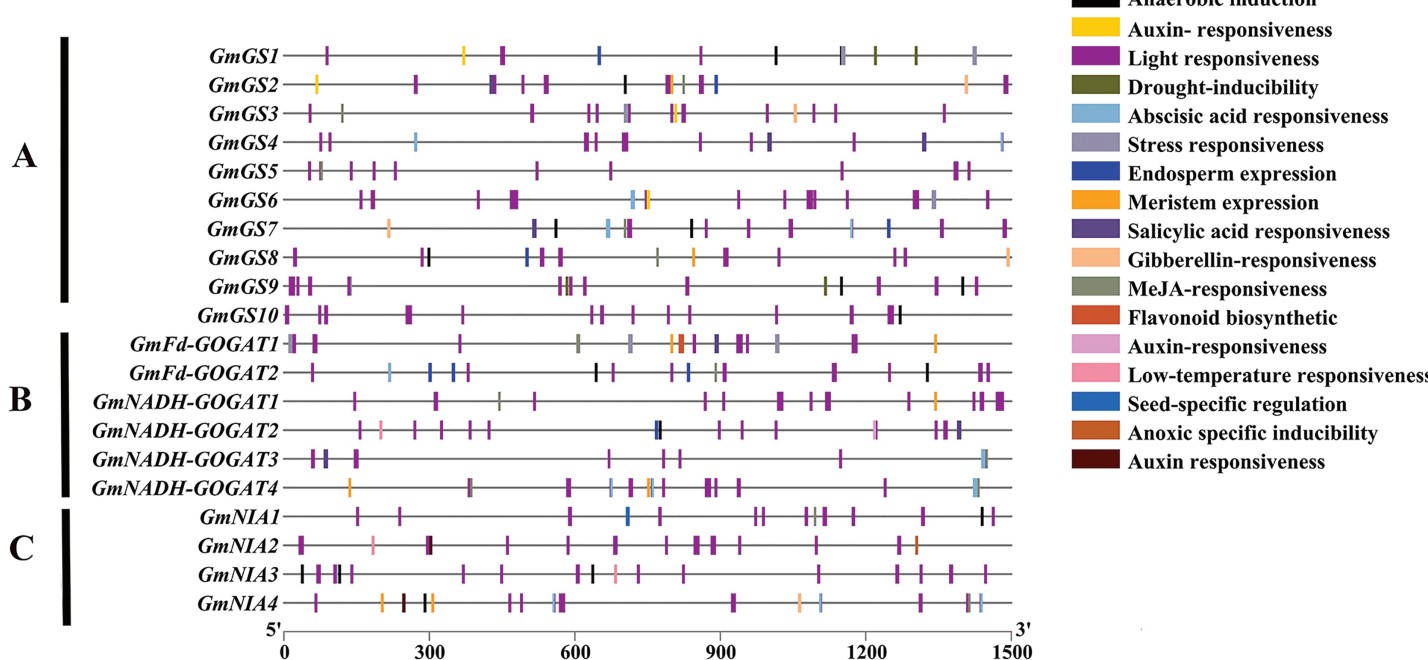

**Figure 5** **Cis-element analysis on the promoter region of *GmGS* (A) *GmGOGAT* (B), and *GmNR* (C).** The potential cis-regulatory elements in the 1,500 bp promoter regions were predicted by PlantCARE software. The elements related to different functional categories were represented by different colors.

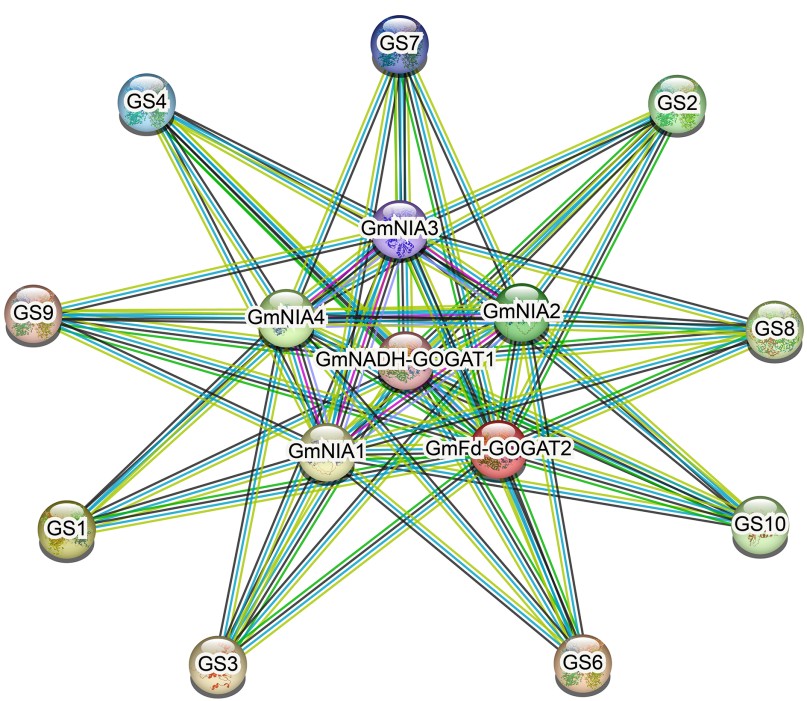

**Figure 6** **The protein-protein interaction network for soybean NAGs.** Various active interaction sources were indicated by different line colors: blue for databases, pink for experiments, green for neighborhood, yellow for text mining, black for co-expression, and blue for protein homology.

## Protein-protein interaction network

As shown in Fig. 6, NAGs engage in interactions with one another. The most effective interaction was observed between *GmGSs* and *GmNRs*. PPI enrichment *p*-value <1.0e−16 indicates that the proteins are at least partially biologically connected, as a group. The potential interactions among NAGs could offer valuable insights for investigating their biological roles.

## Tissue-specific expression profiles of soybean NAGs

Transcriptomic data from the Phytozome database was utilized to investigate the constant expression of the *GmGS*, *GmGOGAT*, and *GmNR* genes in the plant tissues. The resulting expression data of *GmGS*, *GmGOAGT*, and *GmNR* were log-transformed and visualized in a heatmap. Among the genes examined, *GmFd-GOGAT1*, *GmGS9*, *GmGS10*, and *GmNIA1* showed relatively distinct high expression patterns across all eight tissues, suggesting potential involvement in the vegetative organs of *G. max* (Fig. 7). In addition, *GmNADH-GOGAT1* exhibits a unique expression pattern with significant tissue specificity, primarily in root regions such as root tip, lateral root, and flowers (Fig. 7B). In contrast, its expression level is significantly lower in the stem, shoot, and leaf. On the other hand, certain genes like *GmNADH-GOGAT2* and *GmNIA3* displayed tissues-specific high expression levels in the stem and leaf, respectively. These results highlight tissue-specific regulation and potential functional roles of these genes in soybean.

## Analysis of GS, GOGAT, and NR gene expression under abiotic and nitrogen stresses

The study aimed to investigate the response of seven soybean NAGs to nitrate, salt, and nitrate-drought stress conditions using qRT-PCR. Results showed that *GmNADH-GOGAT3*, *GmGS4*, and *GmGS6* did not exhibit significant changes in transcript levels under nitrate stress, while *GmGS10* was significantly induced under both high and low nitrate treatments. These results suggest that NAGs in soybean may play a potential role in responding to nitrate stress (Fig. 8A). *GmNADH-GOGAT1* and *GmGS4* were significantly up-regulated in response to salt stress over time. In addition, *GmNADH-GOGAT3* showed time-dependent regulation patterns, which were initially significantly up-regulated at 24 h and later down-regulated at 48 h, suggesting a possible time-dependent regulation mechanism in response to salt stress (Fig. 8B).

Under drought-nitrate stress treatments (D-HN, D-NN, and D-LN), the expression pattern of these genes was complex (Fig. 8C). For example, *GmNADH-GOGAT1*, *GmGS4*, *GS6*, and *GmNIA2* were significantly down-regulated in response to all drought-nitrate treatments. Additionally, *GmNADH-GOGAT3* was down-regulated under D-HN treatment but up-regulated under D-NN treatment. Interestingly, *GmFd-GOGAT1* and *GmGS10* were significantly up-regulated after all drought-nitrate treatments compared to the control, suggesting a diverse stress response mechanism among NAG genes in soyabean. Overall, these findings highlight the crucial role of NAGs in nitrogen and abiotic stress responses in soybean.

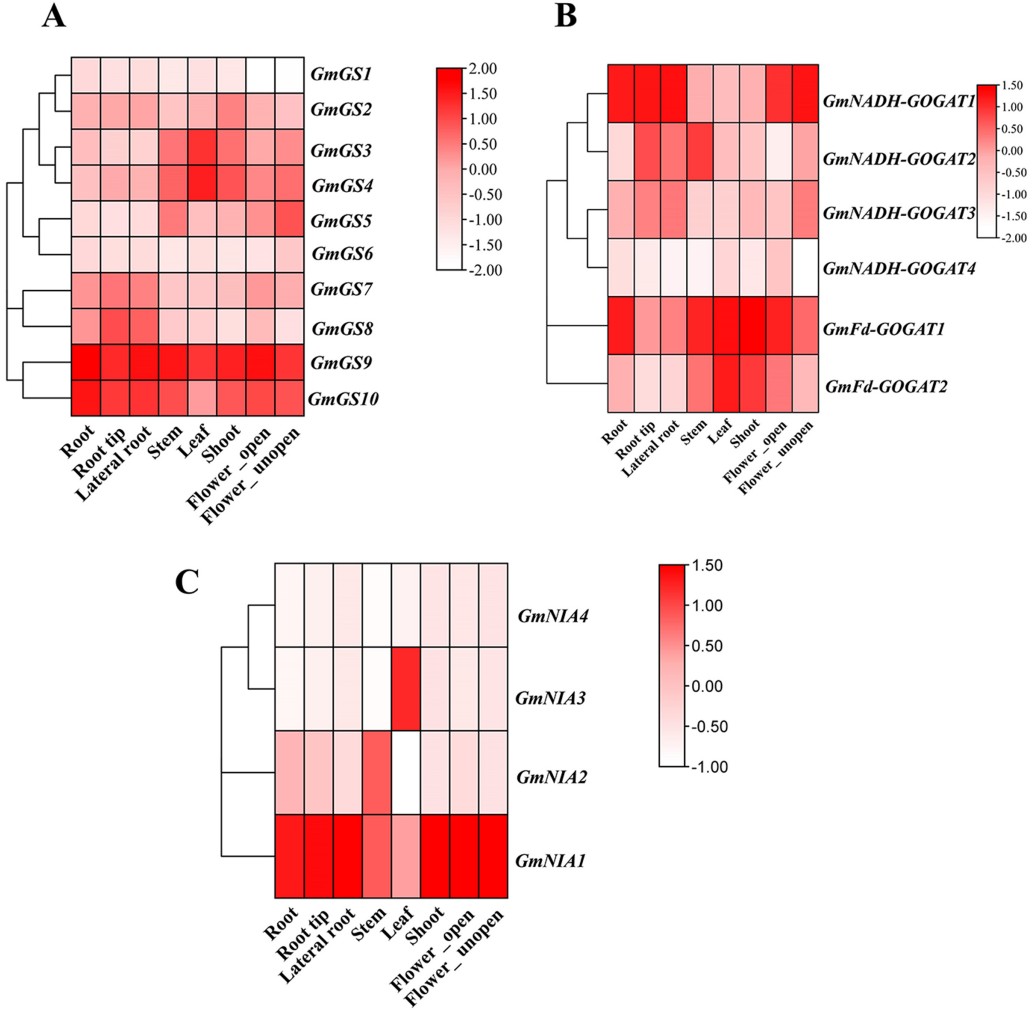

**Figure 7 Expression patterns of *GmGS* (A), *GmGOGAT* (B), and *GmNR* (C) genes in eight soybean tissues.** RNA-Seq data were used to construct the expression patterns using the FPKM values. The color scale bars on the right represent the gene expression levels.

## DISCUSSION

The *GS*, *GOGAT*, and *NR*, which are among the most crucial NAGs, have been confirmed to be involved in various biological processes, including plant stress tolerance (*He et al., 2021*; *Li et al., 2020*). While these gene families have been extensively studied in several plant species, knowledge of their functions in soybean remains limited (*Liang et al., 2022*; *Wang et al., 2021*). In this study, 10 *GS*, 6 *GOGAT*, and 4 *NR* genes were identified and characterized through a comprehensive analysis of the soybean genome. We also investigated their phylogeny, duplication patterns, protein sequences, and expression profiles under nitrogen, drought-nitrogen, and salt stress conditions. A comparative phylogenetic analysis revealed a high degree of conservation in NAGs across various legume species including *G. max*, emphasizing the close relationship among nitrogen assimilation genes in diverse crops. The classification of NAGs was consistent with gene structures, motif distributions, and existing literature (*Li et al., 2022*; *Qiao et al., 2023*).

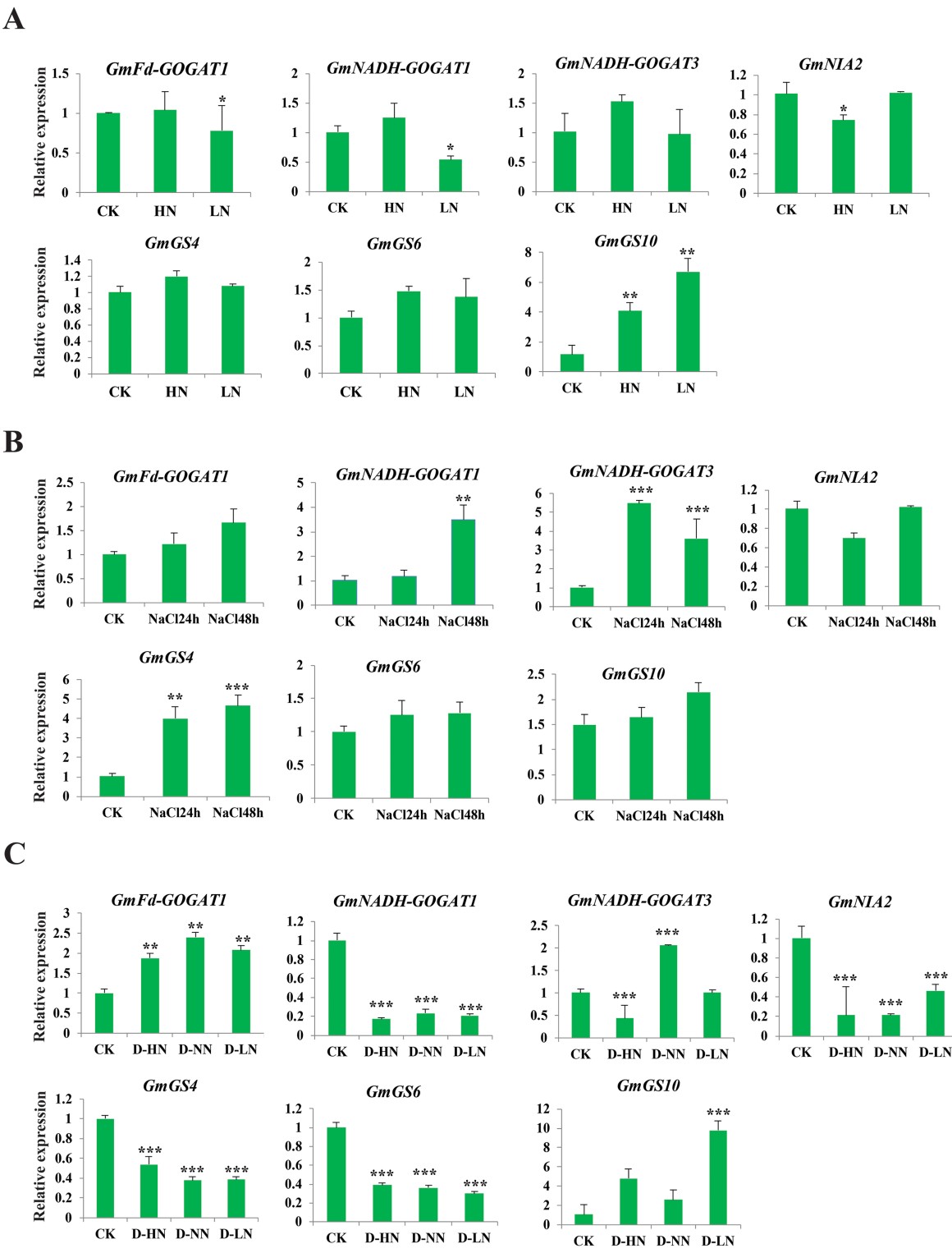

**Figure 8 Expression profile of seven selected *GmGS, GmGOGAT,* and *GmNR* genes in response to various stress treatments.** The x axis labeling the stress treatments and y axis labeling the relative expression. (A) Different nitrate concentrations. (B) A total of 24 and 48 h NaCl stress. (C) Drought-nitrate stress. Gene expression levels were determined using qRT-PCR and normalized with soybean *Actin11* (Glyma18g290800) as a reference gene. Statistically significant expression differences were identified using t-test, *** denotes $p < 0.001$, ** denotes $p < 0.01$, * denotes $p < 0.05$.

Tandem and segmental duplications are thought to have played a significant role in the expansion of gene families over course of evolution (*Jiang, González & Ramachandran, 2013*). The studies in *B. napus* has shown that the expansion of the NAGs family was mainly driven by segmental duplication (*He et al., 2021*). These findings are in line with our current study. A majority of duplicated NAGs pairs were identified as a result of segmental duplication, indicating that it was the primary mechanism driving the expansion of NAGs in soybean during evolution (Fig.2; Table S2). Additional examination of the evolutionary selective pressure revealed that NAGs experienced a strong purifying selection during evolution, implying that their functions may have been conserved over time (Table S2).

The analysis of exon-intron organization and motif patterns within gene families can provide valuable insights into evolutionary relationships (*Liu et al., 2023*). In the current study, the gene structure and motif analysis revealed that genes within the same group tended to have a similar number of introns, similar intron phases, and shared conserved motifs, indicating a pattern of clustering based on these features and strongly supporting the results of the phylogenetic analysis. This observation suggests that these genes share common functions and have evolved from a common ancestor (*Li et al., 2017*; *Wang et al., 2024*).

The interaction between RNA polymerase and the promoter is a crucial event at the onset of transcription, a crucial process in gene expression. The structure of the promoter influences both the binding affinity of the RNA polymerase and gene expression level (*Shang et al., 2013*; *Wang et al., 2024*). The analysis of cis-acting elements in the promoter regions of the *GS*, *GOGAT*, and *NR* genes in soybean provides valuable insights into the potential regulatory mechanisms of these genes. Numerous cis-elements related to light responsiveness were identified, suggesting that these genes might be light-regulated and in plant growth and development processes that are influenced by light conditions. Previous studies have highlighted the role of light in regulating nitrogen accumulation (*Liang et al., 2022*). Additionally, hormone responses and stress tolerance cis-elements were found in soybean NAGs. This implies a role for the soybean NAGs in modulating hormone response and stress tolerance. Our observations align with previous research on rice and pecan (*Qiao et al., 2023*; *Rohilla & Yadav, 2023*).

Moreover, we assessed the expression patterns of *GS*, *GOGAT*, and *NR* genes in different tissues, such as root, stem, and flower, using RNA-seq data. The results revealed that these NAGs have a wide range of expression across these tissues, with *GmGS6* exhibiting the highest expression levels across all tissues. Previous studies have reported the widespread expression of NAG family members in various tissues and organs, indicating their involvement in regulating plant growth and development (*Wang et al., 2021*).

Furthermore, the expression patterns of NAGs were validated using RT-qPCR (Fig. 8), indicating that the NAGs might play a crucial role in responding to a wide range of abiotic stresses and contributing to the development of resistance mechanisms, aligning with previous study (*Li et al., 2022*). The expression of selected *GmGS*, *GmGOGAT*, and *GmNR* genes in response to different nitrate treatments highlights their potential crucial role in

soybean plants' nitrate response. Specifically, under high nitrate (HN) treatment conditions, all selected NAGs showed up-regulated expression patterns, aligning with the results of *Balotf, Kavoosi & Kholdebarin (2016)*, who observed up regulation of wheat NAG expression in response to high nitrate (50 mM $KNO_3$) treatment. Similarly, *Qiao et al. (2023)* reported significant up regulation of pecan *GS* genes under high nitrate concentrations. These studies collectively suggest that these genes may play a vital role in the response of plants to high nitrate levels, potentially through their involvement in the assimilation and metabolism of nitrogen compounds. In contrast, the NAGs showed complex expression under low nitrate treatments (6 mM $KNO_3$); *GmGSs* were up-regulated while *GmGOGAT*s were down-regulated, similar to studies in *B. napus*. The observed different expression patterns of NAGs genes under different nitrogen treatments suggest that these genes have distinct reactions and regulatory mechanisms in response to varying nitrogen treatment conditions. However, the precise mechanisms underlying these differential gene expression patterns and their implications for plant adaptation to nitrate stress require further elucidation. This would contribute to our understanding of plant stress responses and may facilitate the development of strategies for enhancing crop resilience to nitrate stress. Given this, we can make a hypothesis that up-regulated genes are positively regulated and down-regulated genes are negatively regulated under different stresses.

The analysis of gene expression under salt and drought-nitrate stress conditions reveals interesting patterns and potential mechanisms of plant response to these environmental stresses. The majority of the selected genes exhibited increased expression under salt stress over time, which aligns with previous research highlighting the up regulation of *GS*, *GOGAT*, and *NR* gene in rice and *Arabidopsis* in responses to salt stress (*Li et al., 2022*; *Tang et al., 2022*). In contrast, the response of NAGs to drought-nitrate stress appears to be more complex. The significant down-regulation of *GmNADH-GOGAT1, GmNIA2, GmGS4, and GmGS10* under drought-nitrate treatments is suggestive of a common stress response mechanism. Drought stress can affect the uptake and transport of nitrate in plants. Under drought conditions, the expression of nitrate transporters may be down regulated, reducing the availability of nitrate. This can further contribute to the decrease in GS, NR, and GOGAT activity observed during drought-nitrate interaction treatment. However, the up-regulation of *GmFd-GOGAT1* and *GmGS10* under all drought-nitrate treatments is particularly intriguing. This deviates from the general trend observed in other genes and hints at a unique stress response mechanism. This is a novel finding that has not been reported in previous studies and needs further investigation.

## CONCLUSIONS

The study analyzed the *GS*, *GOGAT*, and *NR* genes in *Glycine max*, including their phylogenetic relationships, chromosomal distribution, gene structure, and tissue-specific expression. The study also explored the interaction of the nitrogen pathway with abiotic stresses using RT-qPCR under nitrogen, salt, and drought-nitrogen stresses. The findings suggest these genes play a crucial role in nitrogen metabolism and are significantly influenced by drought and salt stresses. The differential expression of these genes under

different stress conditions suggests their potential role in stress tolerance mechanisms. This research opens new avenues for understanding the complex interplay between nitrogen metabolism and stress response in plants, with further studies needed to understand the precise regulatory mechanisms and explore the potential of these genes in improving drought, salt, and nitrate stress tolerance in crops.

## LIST OF ABBREVIATIONS

| | |
|---|---|
| **Fd-GOGAT** | Ferredoxin glutamate synthase |
| **FPKM** | Fragments per kilobase of transcript per Million mapped reads |
| **GS** | Glutamine synthetase |
| **GOGAT** | Glutamate synthase |
| **NRT** | Nitrate transporters |
| **ATM** | Ammonium transporters |
| **ROS** | Reactive oxygen species |
| **MW** | Molecular weight |
| **MS** | Murashige and Skoog liquid medium |
| **NADH-GOGAT** | Nicotinamide adenine dinucleotide (NAD) + hydrogen (H) glutamate synthetase |
| **NAG** | Nitrogen assimilation gene |
| **NR** | Nitrate reductase |
| **NUE** | Nitrogen use efficiency |
| **qRT-PCR** | Quantitative real-time polymerase chain reaction |
| **II** | Instability index |
| **pI** | Isoelectric point |

### Funding

Funding was provided by the Natural Science Foundation of Jiangsu Higher Education Institutions of China, Grants Number 23KJA210003, and the Open Project Program of Joint International Research Laboratory of Agri-culture and Agri-Product Safety, the Ministry of Education of China, Yangzhou University, Grant Number JILAR-KF202202. The funders had no role in study design, data collection and analysis, decision to publish, or preparation of the manuscript.

### Grant Disclosures

The following grant information was disclosed by the authors:
The Natural Science Foundation of Jiangsu Higher Education Institutions of China: 23KJA210003.
The open Project Program of Joint International Research Laboratory of Agri-culture and Agri-Product Safety, the Ministry of Education of China, Yangzhou University: JILAR-KF202202.

## Competing Interests

The authors declare that they have no competing interests.

## Author Contributions

- Hind Abdelmonim Elsanosi conceived and designed the experiments, performed the experiments, analyzed the data, prepared figures and/or tables, and approved the final draft.
- Tiantian Zhu performed the experiments, prepared figures and/or tables, and approved the final draft.
- Guisheng Zhou analyzed the data, authored or reviewed drafts of the article, and approved the final draft.
- Li Song conceived and designed the experiments, analyzed the data, authored or reviewed drafts of the article, and approved the final draft.

## Data Availability

The raw data is available in the Supplemental File.

## Supplemental Information

Supplemental information for this article can be found online at http://dx.doi.org/10.7717/peerj.17590#supplemental-information.

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
