# Peer review of "Genomic organization and expression profiles of nitrogen assimilation genes in Glycine max"

_PeerJ, doi:10.7717/peerj.17590_

## Round 0.1 · original submission · Major Revisions

Dear Dr. Song,

Thank you for your submission to PeerJ.

It is my opinion as the Academic Editor for your article - Genomic organization and expression profiles of nitrogen assimilation genes in Glycine max - that it requires several major and minor changes to make it suitable for publication.

The reviewers have pointed out a number of shortcomings that need to be adequately addressed to further improve the quality of the manuscript. Therefore, you are advised to carefully go through each and every comment and suggestions, and modify the manuscript.

It is pertinent to mention that your revised submission will undergo additional review to ensure that you have appropriately replied to the reviewers comments.

Hope to receive the revised manuscript in due course.

Reviewer 1 ·

Basic reporting

The manuscript “Genomic organization and expression profiles of nitrogen assimilation genes in Glycine max” has been reviewed critically. The study chosen was most appropriate in the context of climate change, and the leads generated from this study help to advance our understanding of nitrogen assimilation genes.
Introduction
• For the first time use complete names instead of abbreviations like NAGs
• Please follow journal guidelines for citing references in the text and reference section as well.
• There is a discussion needed for nitrogen metabolism in the introduction
• The roles of Glutamine synthetase (GS), Glutamate synthase (GOGAT), and Nitrate reductase (NR) need to be elaborated. Here, at present, their location was described rather than their functional role.
• The impact of Soil salinity, drought, and high temperatures on soybeans needs to be defined qualitatively and quantitatively
• Line # 72-73 Genus names must be written completely for the first time

Experimental design

• Can you provide the photographs of the seedling before and after stress induction
• Line #143-144: If it is a standard protocol, a suitable reference may be cited. If it is novel, why the authors chose those concentrations of nitrate needs to be justified.
• What are the housekeeping genes used in expression analysis
• Can you provide the pictures of the expression analysis carried out

Validity of the findings

The study is novel.
All the data is provided
Logical interpretations of the results obtained in the study were made.

·

Basic reporting

The chosen topic for the study was basic/ molecular research aids in understanding the genes involved in the nitrogen assimilation process under different stress conditions in soybean. The background and introduction were narrated nicely but need to address some minor grammatical and typo errors and also strengthened by citing yield or economic impacts to highlight the problem.

Experimental design

Experimental methodology was appropriate explaining each and every bioinformatic tools, technique and related software used in the study. However, the authors ensure to provide citations or explain about imposition of treatments in brief (as mentioned in the corrected MS). Kindly refer attached document for further minor corrections and rectify them.

Validity of the findings

The results are discussed adequately. However, spelling or typo errors to be rectified. Re-check some of the statements in the MS and data/graph to match correct observations. Emphasize more on specific findings which are very significant in terms of information to be deciphered so that others can use the basic knowledge in applied research in the discussion end or conclusion section.

Additional comments

Re-check some of the statements in the MS and data/graph to match correct observations. Include proper labeling in graph/figures. Rectify grammatical and typing errors.

·

Basic reporting

Comments to the Author

The manuscript titled “Genomic organization and expression profiles of nitrogen assimilation genes in Glycine max” by Elsanosi et al. deals with the identification and analysis of genes associated with nitrogen metabolism in six legume species, using bioinformatics. Furthermore, this paper contributes to the understanding of the interaction of nitrogen metabolism with the response to abiotic stress in crops. Overall, the study has scientific importance, however few corrections still need to be made.

Minor problems:

Line 16: Replace the “in plants .” to “in plants.” and “However ,” to “However,”

Lines 25: I think it would be interesting to describe the meaning of NAGs.

Line 33-34: The sentence is confused. I recommend reviewing it.

Line 37: Add a space to start the paragraph.

Line 48: Replace the “cycle [6, 7]. .” to “cycle [6, 7].”

Line 50: Add a space between the word cytoplasm and the reference.

Line 61: Adjust the space after NO3-

Line 99: “G. max.” instead of “G.max”

Line 111: Replace “V13” to “v13”

Line 115: I think the authors should start the sentence with “In order to” instead of “To”

Line 121: “G. max.” instead of “G.max”

Line 124: Replace the “[27].The” to “[27]. The”

Line 134: Connect this sentence with the previous sentence.

Line 135: Replace “Cytoscape V3.10.1” to “Cytoscape v3.10.1”

Line 167: Adjust the space after the plus symbol.

Line 198: Add a space between the word “Chromosome” and the number “14”.

Line 202: Adjust the space “(Fig.2).” to “(Fig. 2).”

Line 206: The same happens here. Adjust the space “(Fig.S1).” to “(Fig. S1).”

Lines 207-208: The sentence is confused. I suggest organizing it better.

Line 219: Replace “level. (Table S5).” to “level (Table S5).”

Line 223: Put the gene name in italics: “GmGS genes.”

Line 236: Replace “Fig3B” to “Fig. 3B”.

Line 236: Replace “GOAGT” to “GOGAT”.

Line 238: Replace “Figr.3C” to “Fig. 3C”.

Line 243: I think the authors should start the sentence with “In order to” instead of “To”

Line 245: Put the gene name in italics: “GmGS genes.”

Line 256: Remove the extra dot after the word “elements. .”

Line 267: Add bold to the first letter of the name “Tissue-specific”.

Lines 269 and 271: Replace “GmGOAGT” to “GmGOGAT”.

Line 306: “G. max.” instead of “G.max”.

Line 349: Add space after reference [21], before the word “reported”.

Line 354: Adjust the space after the gene name “GmGOGAT s”

Line 361: I think it would be better to start the sentence with “Given this” instead of “From the above”.

Figures 1, 2, 3 and 5: I think it would be better to put the gene names in italics.

Figure 4: Replace “G.max” to “G. max”; replace “C.arietinum” to “C. arietinum”.

Experimental design

No comment

Validity of the findings

No comment

Additional comments

No comments

---

## Round 0.2 · accepted · Accept

Dear Dr. Song,

Thank you for your submission to PeerJ.

I am writing to inform you that your manuscript - Genomic organization and expression profiles of nitrogen assimilation genes in Glycine max - has been Accepted for publication. Congratulations!

Reviewer 1 ·

Basic reporting

It is very clear and unambiguous, significant improvements have been made in the revsion

Experimental design

Appropraite citations were made as per suggestions.

Validity of the findings

Results are valid since they have carried out expression studies too.

Additional comments

References may be corrected as per journal format.

·

Basic reporting

The authors included all the suggestions.

Experimental design

Corrected manuscript as per suggestions.

Validity of the findings

Revised as per suggestions and MS was improved.

Additional comments

The authors attempted all the suggested comments and included them in the revised MS.